# Age-Period-Cohort Analysis on the Time Trend of Hepatitis B Incidence in Four Prefectures of Southern Xinjiang, China from 2005 to 2017

**DOI:** 10.3390/ijerph16203886

**Published:** 2019-10-14

**Authors:** Weidong Ji, Na Xie, Daihai He, Weiming Wang, Hui Li, Kai Wang

**Affiliations:** 1College of Public Health, Xinjiang Medical University, Urumqi 830011, China; m13139661763@163.com; 2Xinjiang Center for Disease Control and Prevention, Urumqi 830054, China; xiena371@163.com; 3Department of Applied Mathematics, Hong Kong Polytechnic University, Hong Kong, China; daihai.he@polyu.edu.hk; 4School of Mathematics Science, Huaiyin Normal University, Huaian 223300 China; weimingwang2003@163.com; 5Central Laboratory of Xinjiang Medical University, Urumqi 830011, China; huihui922@126.com; 6Department of Medical Engineering and Technology, Xinjiang Medical University, Urumqi 830011, China

**Keywords:** HB, age effect, period effect, cohort effect, prediction

## Abstract

**Objective**: The influence of age, period, and cohort on Hepatitis B (HB) incidence in four prefectures of southern Xinjiang, China is still not clear. This paper aims to analyze the long-term trend of the HB incidence in four prefectures of southern Xinjiang, China and to estimate the independent impact of age, period and cohort, as well as to predict the development trend of HB incidence in male and female groups, then to identify the targeted population for HB screening by the model fitting and prediction. **Method**: The data were from the Case List of HB Cases Reported in the Infectious Disease Reporting Information Management System and the Xinjiang Statistical Yearbook of China. The age-period-cohort (APC) model was used to estimate the impacts of age, period and cohort on HB incidence, which could be used to predict the HB incidence in specific age groups of men and women. **Results**: Under the influence of age effect, the incidence of HB in males had two peaks (20–35 years old and 60–80 years old), the influence of age effect on the incidence of HB in females was lower than that of males and the obvious peak was between 20–30 years old; the period effect on the HB incidence in males and females fluctuated greatly and the fluctuation degree of influence on males was bigger than that of women. The HB incidence among males and females in the four regions tended to be affected by cohort effect, which reached a peak after 1990 and then declined sharply and gradually became stabilized. By predicting the HB incidence from 2018 to 2022, we found that there were significant differences in HB incidence among people over 35 years old, under 35 years old and the whole population in four prefectures of southern Xinjiang, China. **Conclusions**: Although the incidence of HB in some regions shows a downward trend, there is still an obvious upward trend of incidences in other places. In our paper, results indicate that the burden of HB incidence may be extended in the future, so we hope this can draw the attention of relative departments. These results reveal the differences of incidence between males and females as well, so respective measures of the two groups’ functions are essential.

## 1. Introduction

Hepatitis B (HB) is an inflammatory lesion of the liver, mainly caused by hepatitis B virus (HBV), which is a kind of infectious disease. Around the world, HBV is one of the causes of liver diseases. HBV can be transmitted by parenteral, sexual, vertical and horizontal routes. Thus, individuals who engage in risky behaviors, such as the use of psychoactive substances, unprotected sex, multiple sexual partners, early initiation of sexual activity and sharing of personal objects, are more vulnerable to HBV infection [1]. According to the official website of the World Health Organization (WHO), one third of the world’s population-about 2 billion people-has been infected with HBV [2]. In 2015, about 887,000 people died of infection with the HB virus. In some areas, the number of deaths caused by viral hepatitis has exceeded the total number of deaths caused by AIDS, tuberculosis and malaria [3]. In the past 50 years, HBV infection has shown moderate or high prevalence level in low-income countries. Although the incidence of acute HB and the prevalence of chronic carriers of HB surface antigen (HBsAg) have been reduced in some countries due to the universal vaccination program for HBV launched in the 1990s, it is still difficult for some countries to implement this program, especially in rural areas where HB is highly prevalent [4].

China is a high prevalence area of HBV. According to the results of the national seroepidemiological survey of HB in 2006, it is estimated that there are 93 million carriers of HBV and 25 million chronic HB patients [5]. Every year, more than 300,000 cases of liver cirrhosis and hepatocellular carcinoma caused by infecting HBV die and there are about 500,000 to 1 million new cases of HB. HB is one of the three most important infectious diseases in China. It is a serious public health problem that endangers people’s health, hinders social development and affects social stability. By 2014, the prevalence of HB has decreased significantly, after the implementation of comprehensive prevention and control measures mainly based on the vaccination of HB vaccine in China since 1992. According to WHO standards, China has generally changed from a high prevalence of HB to a moderate epidemic area. However, the infection rate of HBsAg in the whole population is still high. The prevalence of HB varies from region to region, presenting a mixed state of high, middle and low epidemic areas.

Xinjiang is located in western China, with a relatively backward social and economic level, obvious cultural differences, vast territory and scattered residents, leading to the incidence of HB in Xinjiang ranking first in the country. From 2005 to 2010, the overall incidence of HB in Xinjiang showed an upward trend, and the highest incidence was 236.30/100,000 in 2008. After 2011, the incidence of HB in Xinjiang showed a downward trend, then maintained a relatively stable level, with the lowest incidence in 2018 at 148.54/100,000. In terms of the distribution of prefectures, there are HB cases in all regions of Xinjiang, mainly concentrate in Kashgar, Aksu, Urumqi, and Kizilsu Kirghiz prefectures; when it comes to the age distribution, all age groups all have HB cases, and mainly concentrate in adolescents and adults aged 20–50 years old; from the occupational distribution point of view, reported cases mainly concentrate in farmers, housework and unemployment; about the seasonal distribution, there are new increases every month but no special seasonal distribution characters. There are few studies on the characteristics of HB in Xinjiang. The analysis about the trend of HB incidence could provide important clues for the prevention and control of HB. Therefore, in order to further understand the distribution of HB in southern Xinjiang and explore the long-term trend of the disease, we conducted a series of studies as follows.

Inspired by Wu et al. [6], we collected data from many sources, including: Xinjiang 2010 census data, Xinjiang statistical yearbook information from 2005 to 2017 and HB case information from four prefectures in southern Xinjiang, China. We combined these data to develop an APC model to differentiate the impact of age structure, historical trends and birth cohort on the incidence of HB from an epidemiological perspective. And we predicted the future trend of HB incidence by fitting APC model.

First, by collecting the detailed case information and demographic statistics, we described the age, sex, occupation and other characters of the HB patients in four prefectures of southern Xinjiang, China. Then, we established APC model in terms of the HB cases in the four prefectures, and investigated the population structure of HB cases and the role of historical trend. Last, we predicted the HB incidence trend of the people under 35 years old, over 35 years old and the whole population in future 5 years.

Based on the abundant data and relatively new HB research methods from 2005 to 2017 in many regions of Xinjiang Uygur Autonomous Region, China, we extracted useful information from each case of HB. In particular, we divided all the subjects into two age groups: Under 35 years old and over 35 years old, as well as male and female groups to predict the development trend of HB incidence and we tried to determine the targeted population of HB screening by model fitting and prediction. We expect that the results in this study could provide insights into the the impact age and year on confirmations in different cities and suggestions on surveillance and preventive.

## 2. Method

### 2.1. Data Source

This paper consists of two data source, one is from the four prefectures in southern Xinjiang (Figure 1), and all cases of HB reported in Aksu, Hotan, Kashgar and Kizilsu Kirghiz from 2005 to 2018. Hepatitis B surface antigen positive patients were diagnosed as HB cases. There are 215,413 HB cases, and the cases in four regions are: 63,697 cases in Aksu, 18061 cases in Hotan, 124,486 cases in Kashgar, and 9169 cases in Kizilsu Kirghiz. We browsed, reviewed and downloaded the case sheet of HB reported in the information management system of infectious disease report from 2005 to 2018 according to the date of onset. Case table includes basic information, diagnosis type, diagnosis time and so on. There are only 17 cases(generally less than 0.001%), we deleted cases over 100 years old for convenience on data processing and exploration. And another data source is from annual population data of the four regions in Xinjiang Statistical Yearbook 2005–2017. According to lacking Xinjiang Statistical Yearbook data in 2018, we failed to get the relative data in four prefectures, so the HB data in 2018 were excluded.

Meanwhile, we retrieved the relatively detailed age structure of the four regions (0, 1–4, 5–9, 10–14, ..., 95–99, and over 100) from the data of the Sixth National Population Census of Xinjiang Statistical Bureau in 2010. We applied the detailed age structure of the 2010 census data to the annual data of 2005–2017, combined the two groups of population data, and constructed the complete population age statistics of the four regions from 2005 to 2017.

In order to calculate the exact population of males and females in each age in the four regions from 2005 to 2017, we have carried out the following steps for the population of each region: (i) calculating the relatively detailed age composition ratio of the 2010 census in these four regions; (ii) applying the above-mentioned age group ratio of 2010 to the statistical yearbook of Xinjiang to obtain the population of each detailed age group from 2005 to 2017; (iii) by considering the proportion of each age group, each detailed age group is subdivided to obtain the population of each age group.

### 2.2. Statistics Analysis

Age, birth cohort and calendar period (i.e., historical trend) are the three intrinsic factors that play essential roles on long-term epidemiological study. However, these three factors share perfect linearity (i.e., age + cohort = period), which makes it difficult to identify their sole impacts [7]. Until recently, Age-Period-Cohort (APC) modelling analysis was proposed [8,9], and was commonly used to study the trends of disease-induced confirmations/mortalities [6,10,11,12,13,14].

The APC Model, a statistical method based on the Poisson distribution, is widely applied in epidemiologic, demographical, and sociological fields, which could be used to extract information from cross-sectional data regarding changes of socioeconomic, environment, and lifestyle in the morbidity and mortality risk [15], termed as cohort effect.

We constructed an APC model by setting the age interval of one year (0–82 years) and the calendar interval of one year (2005–2017) to estimate the impact of age, period and birth cohort on the incidence of HB, and to explore the long-term trend of the incidence of HB.

Then the last 18 age groups (83–100) were clustered into a group, which was classified as over 82 years old. We eventually merged the population statistics of HB incidence from 2005 to 2017 and divided these data into 84 age groups for APC models to explore the impact of age, period and birth cohort.

We fitted APC models to the HBV confirmation, and to estimate the sole effect of age, birth cohort and period on HBV confirmation rate. 100 times bootstrapping with Poisson simulation were conducted for sensitivity exploration [16]. We predicted the future populations for both genders by fitting them with smooth splines [17,18]. By applying Vector Auto regression (VAR) method [19] to APC models, a 5-year ahead forecast was conducted for HBV confirmation rate.

All analyses, data visualizations and modeling were programmed in R (version 3.6.0). In particular, R package “*apc*” was implemented for fitting APC models and forecasting the future trends [20].

## 3. Results

Incidences in the four prefectures of southern Xinjiang, China were shown in Table 1. It showed that during the 13 years, there were 195,932 HBV cases, 103,763 were males and 92,169 were females. The number of cases in the each district: 56,468 cases in Aksu area, 16,753 cases in Hotan area, 113,928 cases in Kashgar area and 8783 cases in Kizilsu Kirghiz area.

The annual incidence of HB among males and females, annual population, annual incidence of HB and corresponding change rate in four prefectures of Southern Xinjiang were shown in Figure 2. Generally, larger population would lead to the increasing number of HB cases, but the incidence rate was uncertain. So in the four prefectures, with the most population, there were the most HB cases in Kashgar, while the fewest cases showed up in Kizilsu Kirghiz. Recent years, the number of patients in Aksu and Kashgar showed an upward trend, while the number in Hotan and Kizilsu Kirghiz was relatively stable. During 13 years, the incidence of HB in Kizilsu Kirghiz changed dramatically, fluctuating between 20% and 200%. The incidence of HB in AKsu and Hotan area experienced peak and valley values, the variation ranged from 10% to 90%. The incidence of HB in Kashgar was relatively flat compared with the other three areas, and the change range was between 10% and 50%. In the four regions, Kashgar had the highest incidence of HB among men and women, almost four times as high as Hotan and twice as high as Kizilsu Kirghiz. After 2014, the incidence of HB among men and women in Aksu region continued to rise. The incidence of HB among men in Hotan region increased sharply after a period of stationary period, while the incidence of HB among women continued to decline. The incidence of HB among men in Kashgar region continued to rise, while the incidence of HB among women tended to stabilize. In Kizilsu Kirghiz, the HB incidence of men and women increased slowly after a stable period. The incidence of HB in the four regions changed during this period (that is, from 2015 to 2017), but the rate of change was different.

In the Figure 3 and Figure 4, we could see the total age density of HB among men and women in four regions of Xinjiang from 2005 to 2017 and the annual age density of HB among men and women in four regions from 2015 to 2017, and we found some similarities from the age density maps of four regions. In Figure 3, in four regions, first of all, a large number of HB cases were found among male and female aged 20, and the age density of HB cases was higher in females than in males; secondly, the proportion of males aged 30 to 45 years in Aksu and Kizilsu Kirghiz was higher; thirdly, the HB infection density of male and female under 15 years old in Hotan and Kashgar showed a small fluctuation; fourthly, the incidence of HB in males was higher than that in females after 35 years old; fifthly, the age density of HB in middle-aged men in Aksu and Kizilsu Kirghiz was similar to that in young people, but not in Hotan and Kashgar; Sixthly, we could see in Figure 4, the age-related morbidity density in these four areas showed a clustering trend over the past 13 years, the HB morbidity density of men and women in Aksu area gradually converged to 25–35 years old and 40–60 years old, respectively, while in Hotan, Kashgar and Kizilsu Kirghiz areas, the HB morbidity density in men and women gradually converged to 25–35 years old.

Figure 5 and Figure 6 depicted the incidence of HB among men and women over 35 years of age in four regions of Xinjiang from 2005 to 2017. From 2005 to 2017, the trend of HB prevalence among males and females over 35 years of age in four regions was similar; the incidence of HB among males and females over 35 years of age in Aksu and Kashgar regions was about 2–6 times higher than that in Hotan region; and the incidence of HB among males and females in Hotan was relatively low over the years, the incidence of male and female in Kizilsu Kirghiz area had gradually decreased in recent years, but the incidence of HB among men and women in Aksu and Kashgar remained high in recent years.

In Figure 7, we proposed the incidence of HB in boys and girls in different periods according to the sex and annual in four regions. There are some similarities among the four regions. Firstly, the incidence of HB among boys and girls at different stages in the four regions had slowly declined in recent years and then tended to be flat; secondly, boys and girls at 0-year-old infant stage in each region had similar changes, and the HB incidence of boys and girls at 0-year-old infant stage in Aksu and Kashgar regions reached the peak in 2013–2015, while in Hotan, the incidence reached peak in 2007, when it comes to Kizilsu Kirghiz, the incidence was relatively flat. Thirdly, the annual incidence of HB in boys and girls aged 1–4 in four areas did not change much. Fourthly, the annual incidence of HB in boys and girls aged 5–14 in four areas was similar, they all experienced a sharp decline after the peak in 2009, and then gradually stabilized.

Figure 8 showed the composition of HB incidence among young adults aged 18 to 35 and middle-aged people aged 36 to 60 in four regions of Xinjiang from 2005 to 2017, and Figure 9 showed the age distribution of HB incidence among different occupations in four regions. It could be seen that the composition of HB incidence in four regions and the age distribution of HB incidence in different occupations had some similarities. Firstly, the incidence of HB among young people and middle-aged people in the four regions was heterogeneous, and the incidence of HB among young people was higher than that of middle-aged people in general; secondly, in the HB cases, the proportion of young people in Aksu showed a downward trend, while in Hotan, Kashgar and Kizilsu Kirghiz prefectures showed an upward trend. Thirdly, the number of middle-aged people with HB in Aksu was on the rise in general, in Hotan, Kashgar and Kizilsu Kirghiz it was relatively stable. In the recent 3–4 years, the number of middle-aged people with HB showed a slowly rising trend. Fourthly, in general, the number of farmers infected with HB was the largest, followed by housework and unemployment, cadres and workers, and most of the students were between 5 and 20 years old, most of the retired workers were after 45 years old, and the rest of the occupations were between 18 and 80 years old (Figure 9).

Figure 10 and Figure 11 showed the estimated impact of age, period and birth cohort on the incidence of HB, and the corresponding predictive trends for males and females in four regions of Xinjiang. Because the age density patterns of men and women had their own characteristics, as it was shown in Figure 9, we had established APC models for men and women in each region respectively. From Figure 10, we could see that in recent years, the risk rate of HB infection in males was slightly higher than that in females in four regions. There were similarities between the estimated impact and the predicted trend patterns in these areas: (i) there were two peaks of male morbidity under the influence of age effect (i.e., 20–35 years old, 60–80 years old); (ii) the age impact of male morbidity could be divided into two stages, under 35 years old and over 35 years old; (iii) women were less affected than men in the whole age range, with a significant peak between 20 and 30 years old; (iv) the influence of period effect on morbidity of men and women fluctuated greatly, and men were higher than women. In recent years, male in the four regions and females in Aksu had been severely affected by the period effect, and the incidence had an increasing trend, while the incidence of Hotan females declined. The incidence of Kashgar and Kizilsu Kirghiz females changed steadily; (v) the trend of cohort effect on male and female morbidity in the four regions was similar, reached the peak around 1990, and then declined sharply and stably. At the same time, regional heterogeneity was also observed: (i) there was a significant difference in age effect between men and women in Hotan and Kizilsu Kirghiz between 60 and 80 years old; (ii) Hotan and Kizilsu Kirghiz men had a second age effect peak between 70 and 80 years old; (iii) men and women in Aksu, Hotan and Kizilsu Kirghiz were significantly affected by the period effect from 2006 to 2010. The period effect in Kashgar reached its peak in 2012, and the period effect in Hotan reached its peak again in 2013; (iv) from 2018 to 2022, the incidence of HB among people over 35 and under 35 years old in four regions was significantly different. The incidence of HB in the population above 35 years old was almost 2–3 times as high as the under 35 group, and 2 times higher than that in the general population. The incidence of HB in female population under 35 years old in Hotan and Kizilsu Kirghiz had fewer differences from that in general population. The incidence of HB among men and women in Aksu, Kashgar and Hotan was significantly different between groups aged over 35 and under 35. And the HB incidence in the population under 35 years of age in four districts all had declined; (v) from 2018 to 2022, the predicted trend of HB incidence in Aksu and Kashgar men and women and in Hotan men over 35 years of age had increased, and the incidence of HB in Kashgar men and women and in Hotan women over 35 years of age had also increased.

## 4. Discussion

We applied a relatively novel method to model the trends in HBV confirmations across four areas among population in Xinjiang, China. Compared to a compartmental model within a Markov process, which needed presumptions on key parameters and mass computation capacity, APC models were able to estimate and forecast directly in a short time. The separate estimations on the effects of age, calendar period and birth cohort provided public health department with insights on targeted age groups and historical trends. Though APC models were usually used for chronic diseases [10,11,12,13,14], it was possible to apply them to infectious diseases like HBV and tuberculosis [6,21,22], given that these infectious diseases shared a long-run infection process.

Our results revealed basic information, including the age structure and time trend of HB. The declining trend of HB incidence in some areas might be related to the effectiveness of HB control measures and treatment programs in China [23]. Analysis of the annual incidence of HB among male and female children in four regions from 2005 to 2017 (Figure 6) showed that the sharp decline of the annual incidence of HB among male and female children aged 5–14 in school age in 2009–2010 might be related to a new policy on HB formulated by China in 2009, which stipulated that under 15 years of age without vaccination were required. Children can be vaccinated free of charge [24].

Age was one of the important factors of HB. According to our results, the age effect of the risk of HB in both sexes was significant at the ages of 20–30 and 60–80, which indicated that the risk of HB was higher in young and elderly people. This trend was mainly attributed to the increasing aging of China’s population [25]. The impact of young people aged between 20 and 30 on the annual incidence of HB could be explained that they were caused by some unhealthy living habits such as unprotected sexual behavior between young men and women, multiple sexual partners [1], many contacts, excessive pressure on young people, frequent staying up late, drinking, smoking, and inappropriate work and so on. Compared with younger people older people were more susceptible to diseases, injuries and environmental pollution [26,27]. In addition, in our study, the age effect of the incidence of HB in males was comparable or even higher than that in females. It was speculated that physiological differences between sexes might be a factor [28], but further research was needed.

Period effects usually indicated the changes that had a direct impact on the incidence of HB, such as new diagnoses, improved medical interventions, etc. In this study, time effects had a significant impact on the incidence of HB. In 2006–2010 the period effect on the HB incidence among men and women in Aksu, Hotan and Kizilsu Kirghiz was the most serious, while in Kashgar the incidence peak because of period effect showed up in 2012, and in Hotan it showed up again in 2013.

Overall, the risk of HB incidence decreased throughout the study cohort. The effect of birth cohort was significant before 1990, suggesting that cohorts born before 1990 had a higher risk of HB. After 1990, it gradually reached its peak, then declined rapidly and became stable, which could be explained by the popularization of HB vaccination [23]. This pattern of change in the birth cohort could also be explained by improving nutrition, living conditions and health care services.

The annual analysis of the age density of HB incidence from 2005 to 2017 (Figure 3) showed that the age density of HB incidence in Hotan and Kashgar was relatively stable over the years, while the proportion of middle-aged population in Aksu and Kizilsu Kirghiz gradually appeared in recent years. Based on these results, we suggested that the relevant departments paid more attention to the control scheme of the middle-aged population. The incidence of HB among young and middle-aged people in the four regions (Figure 8) had relatively obvious heterogeneity. The analysis showed that the incidence of HB among young and middle-aged people in the four regions had declined in recent years, while the incidence of HB among middle-aged people had increased. Further studies on the occupations of HB (Figure 9) showed that farmers were the most attacked occupations, followed by housework and unemployed persons, cadres’support and workers. Because of poor living conditions and difficulty in obtaining medical care, people living in rural areas were very vulnerable to common infectious diseases, especially HB [23], and therefore they were vulnerable to various infectious diseases. It suggested that more efforts should be made in preventing and treating HB in rural areas than in urban areas in order to maintain the downward trend of HB incidence.

Our predicted results showed that the total population of Aksu and Kashgar would have an upward trend in the future, and the incidence of HB under 35 years of age would have a downward trend; the HB incidence of the total population in Hotan and people under 35 years old would have a downward trend, and the incidence of HB in men over 35 years of age would have a downward trend, while the HB incidence in people over 35 years old was on the rise; the trend of HB incidence in Kizilsu Kirghiz would decline. Therefore, we suggested that the public health departments in Aksu, Kashgar and Hotan should focus on the upward trend of the above population in order to better control the epidemic of HB. Our results also showed that the incidence of HB was different between males and females. Therefore, different measures should be taken to control the incidence of HB in both sexes.

The main limitation of our study was the rough statistics of population data of age distribution, which might bring small noise and errors to estimation. Therefore, more detailed statistics and censuses were needed to improve estimation and prediction.

## 5. Conclusions

Although the decreasing trend of HB incidence was observed in some areas, the incidence of HB in other areas was still on the rise. As China’s aging population is becoming more and more serious, the young people’s social circle is becoming more and more extensive, and because of being under too much pressure, there are lots of unhealthy living habits such as frequent drinking, smoking, inappropriate work and rest and so on, which might lead to the burden of HB in some areas, so the incidence rate of it might increase in the next few years. From above phenomenon, we hope to attract the local public health departments attention in these areas to focus on the prevention and control of HB. The results also show that the incidence of HB is different between males and females. Therefore, measures to control the incidence of HB should be proposed for both sexes.

## Figures and Tables

**Figure 1 ijerph-16-03886-f001:**
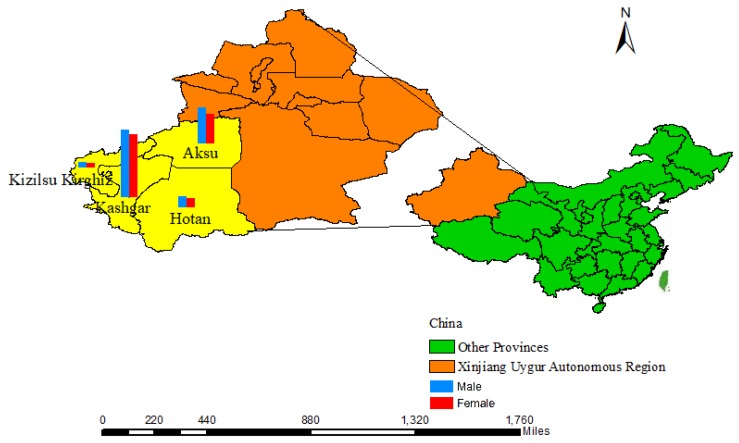
The specific geography location of four prefectures in southern Xinjiang.

**Figure 2 ijerph-16-03886-f002:**
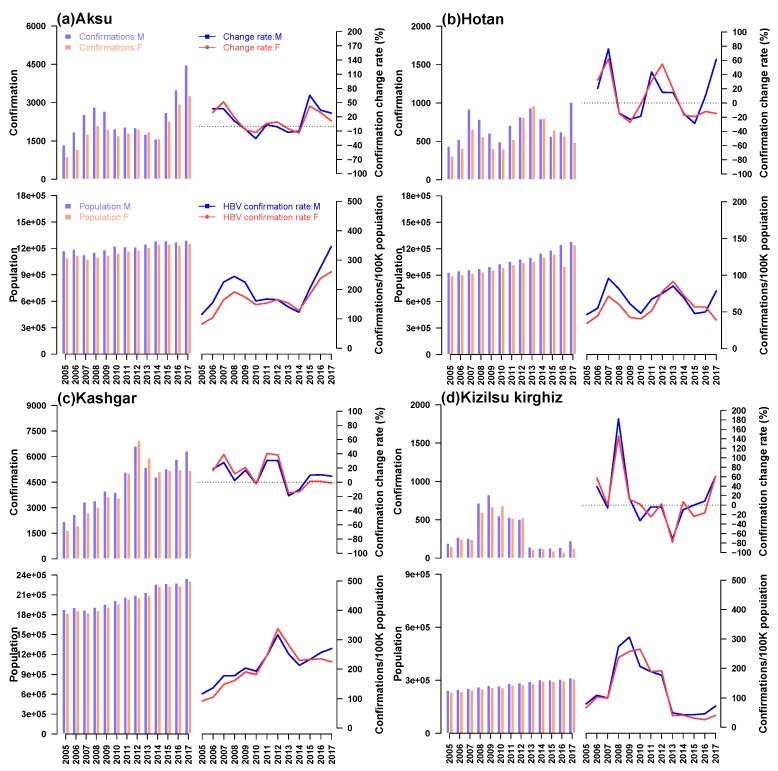
A description of the total population and the population with HBV from 2005 to 2017 in four regions of Xinjiang, China. Blue stripes and lines denoted men, red stripes and lines denoted women. In each group, for men and women, the upper bar chart showed the number of cases of HB, the upper bar showed the change rate of HB incidence, the lower bar chart showed the annual population of the region, and the lower bar showed the incidence rate of HB.

**Figure 3 ijerph-16-03886-f003:**
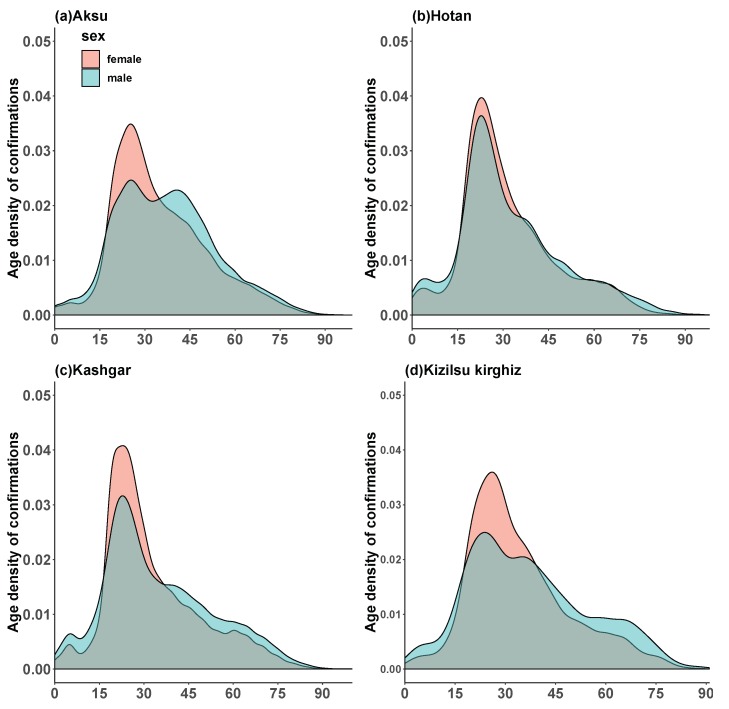
Age densities of HB in women and men in four regions of Xinjiang from 2005 to 2017. In each group, blue and red lines represented the age densities of HB in men and women respectively.

**Figure 4 ijerph-16-03886-f004:**
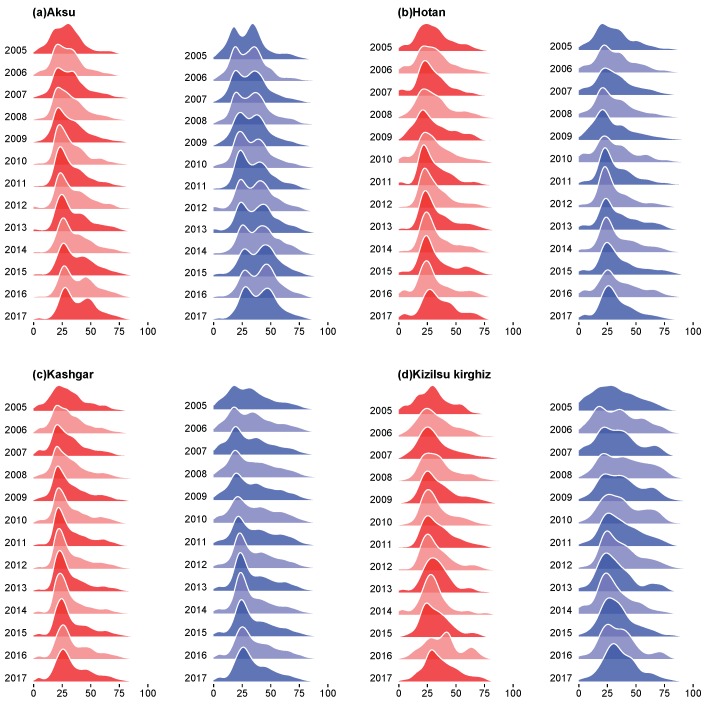
Age structure densities of HB among women and men in four regions of Xinjiang from 2005 to 2017. In each group, red meant female and blue meant male.

**Figure 5 ijerph-16-03886-f005:**
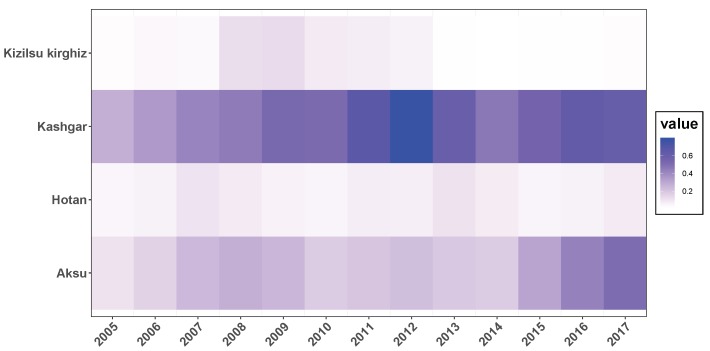
Incidence of HB among men over 35 years of age in four regions of Xinjiang from 2005 to 2017 (%).

**Figure 6 ijerph-16-03886-f006:**
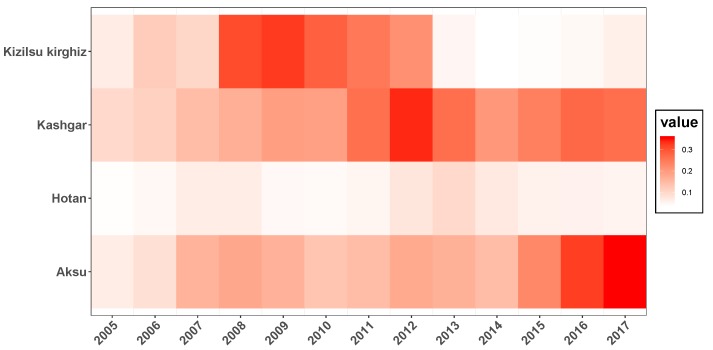
Incidence of HB among women over 35 years of age in four regions of Xinjiang from 2005 to 2017 (%).

**Figure 7 ijerph-16-03886-f007:**
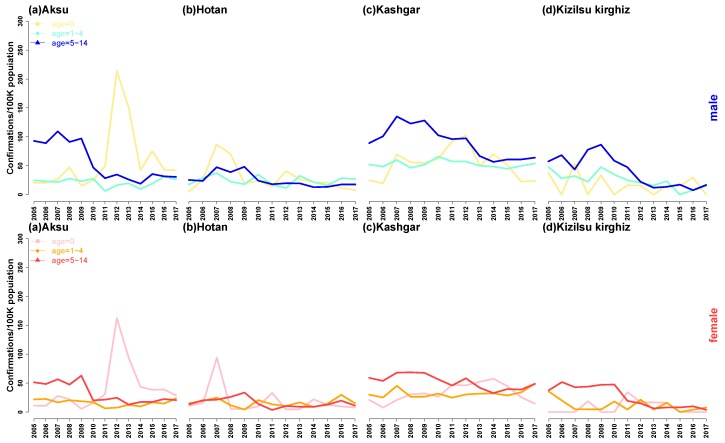
Annual incidence of HB among male and female children in four regions of Xinjiang, China, from 2005 to 2017. In male children, the yellow line indicated 0-year-old infant, the green line indicated 1–4-year-old infant, and the blue line indicated 5–14-year-old school age. In female children, the purple line indicated 0-year-old infant, the orange line indicated 1–4-year-old infant, and the red line indicated 5–14-year-old school age.

**Figure 8 ijerph-16-03886-f008:**
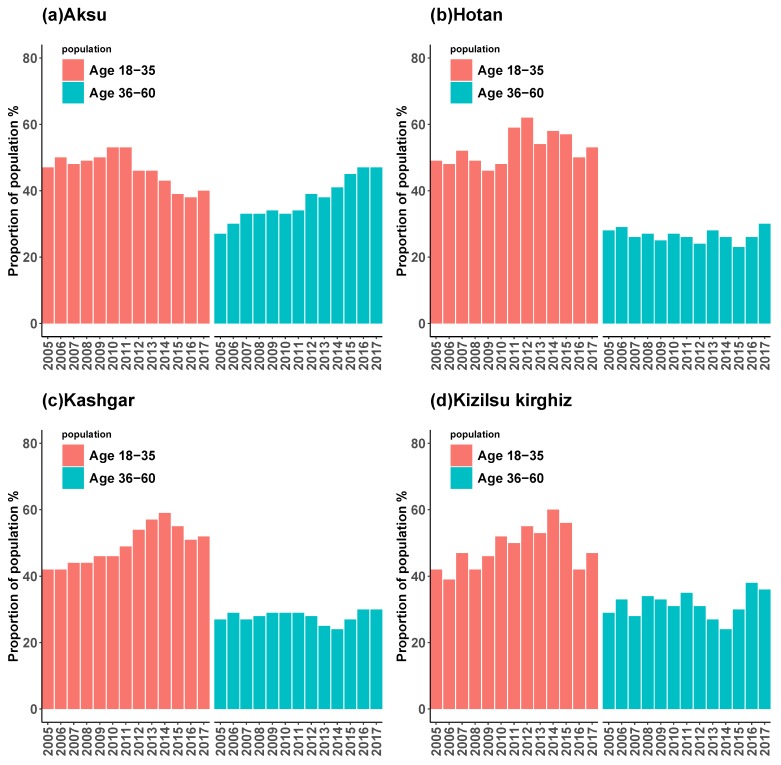
The incidence of HB among young adults aged 18–35 and middle-aged people aged 36–60 years in four regions of Xinjiang, China, from 2005 to 2017 (%). In each group, red meant young people and blue meant middle-aged people.

**Figure 9 ijerph-16-03886-f009:**
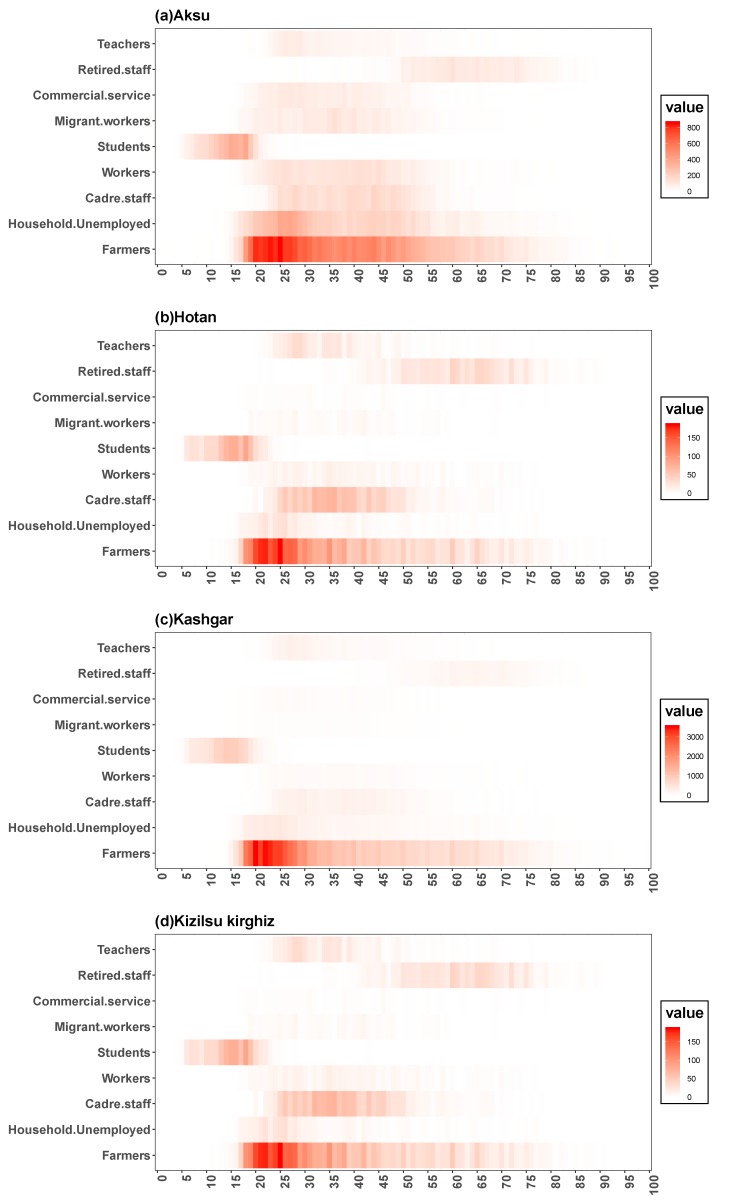
Age distribution of HB incidence in different occupations in four regions of Xinjiang.

**Figure 10 ijerph-16-03886-f010:**
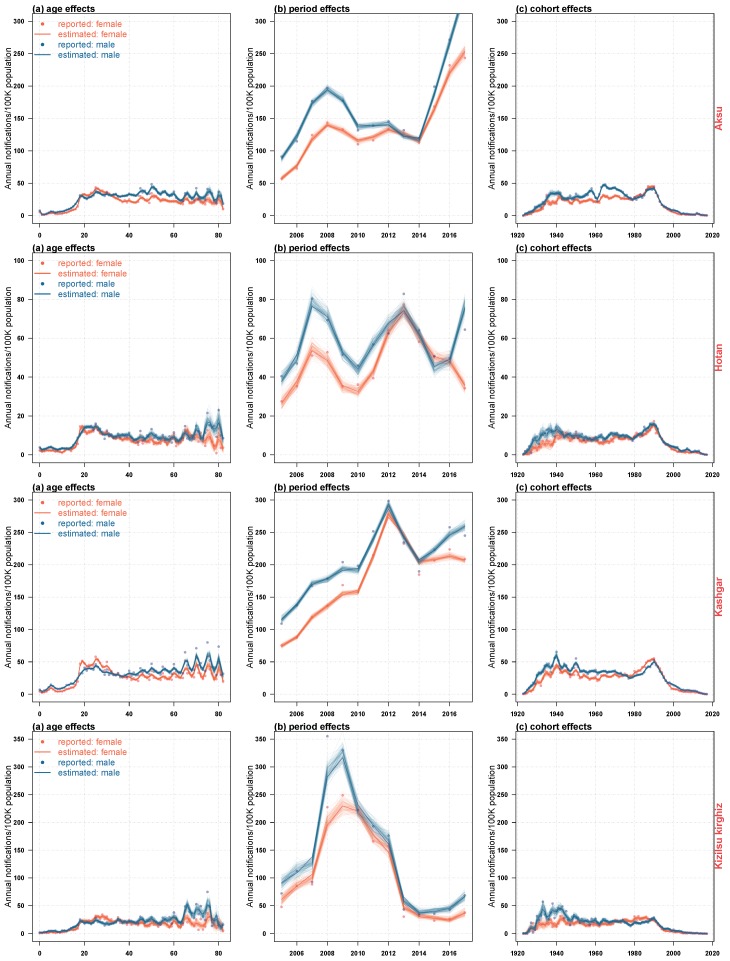
The estimated impact of age (**left column**), period (**middle column**) and birth cohort (**right column**) on the incidence of HB in four regions of Xinjiang from 2005 to 2017.

**Figure 11 ijerph-16-03886-f011:**
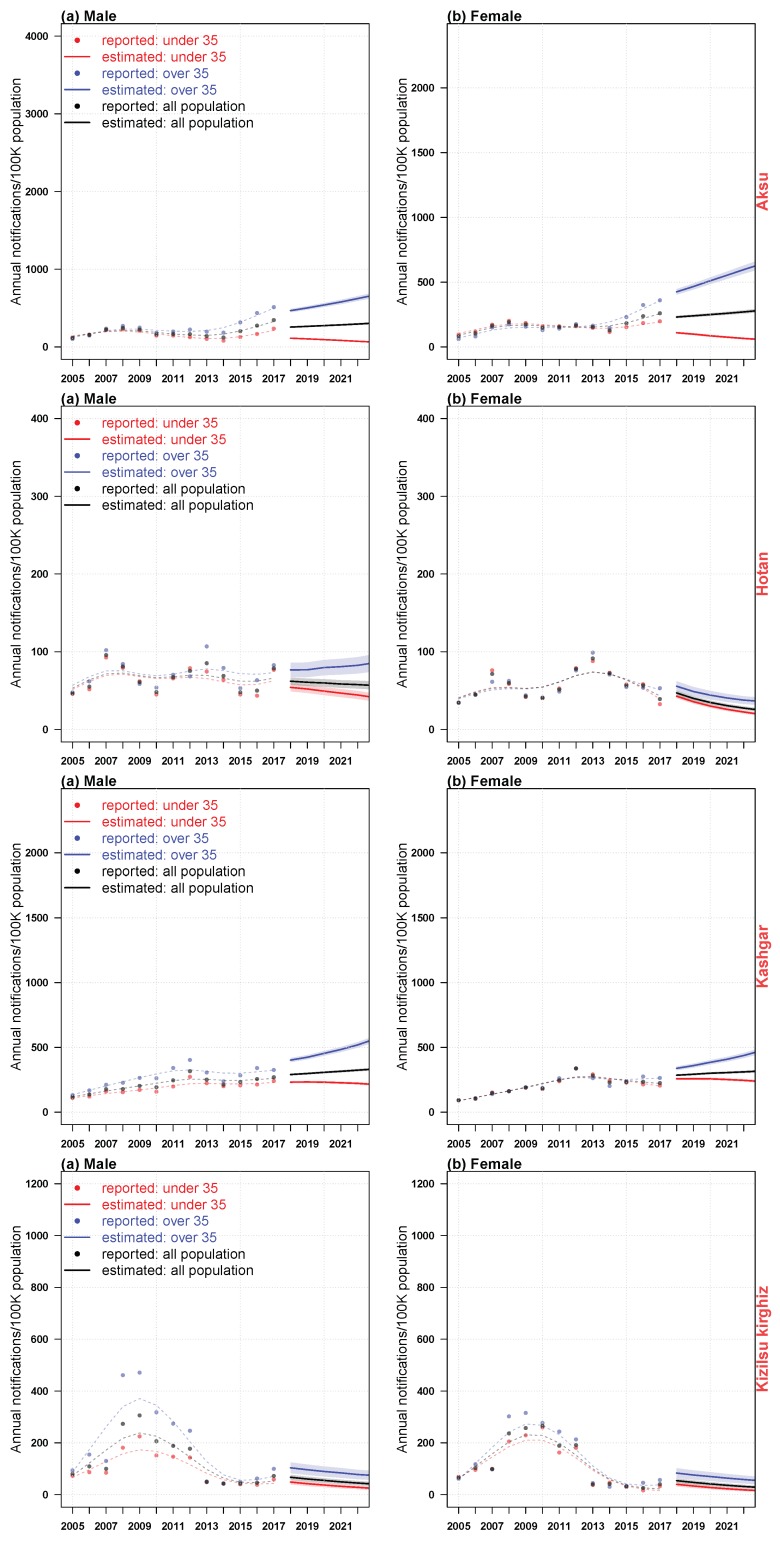
The trend of HB incidence in four regions of Xinjiang from 2005 to 2017, and the prediction of HB incidence trend from 2018 to 2022.

**Table 1 ijerph-16-03886-t001:** Summary of HBV confirmations across four areas in Southern Xinjiang.

	Aksu		Hotan		Kashgar		Kizilsu Kirghiz	
	(N = 56,468)		(N = 16,753)		(N = 113,928)		(N = 8783)	
	no.	% (95%CI)	no.	% (95%CI)	no.	% (95%CI)	no.	% (95%CI)
Gender								
Male	31171	55.20 (54.79–55.61)	9219	55.03 (54.27–55.78)	58,751	51.57 (51.28–51.86)	4622	52.62 (51.57–53.67)
Female	25,297	44.80 (44.39–45.21)	7534	44.97 (44.22–45.73)	55,177	48.43 (48.14–48.72)	4161	47.38 (46.33–48.43)
Age groups								
0–14	2621	4.64 (4.47–4.82)	1435	8.57 (8.15–9.00)	8495	7.46 (7.30–7.61)	517	5.89 (5.40–6.40)
15–29	19,835	35.13 (34.73–35.52)	7537	44.99 (44.23–45.75)	49,326	43.30 (43.01–43.58)	3304	37.62 (36.60–38.64)
30–50	23,682	41.94 (41.53–42.35)	5326	31.79 (31.09–32.50)	35,476	31.14 (30.87–31.41)	3271	37.24 (36.23–38.26)
Over 50	10,330	18.30 (17.98–18.61)	2460	14.68 (14.15–15.23)	20,643	18.12 (17.90–18.34)	1691	19.25 (18.43–20.09)
Occupations								
Farmers	23,141	40.98 (40.57–41.39)	9897	59.08 (58.33–59.82)	67,007	58.82 (58.53–59.10)	3612	41.12 (40.09–42.16)
Household/Unemployed	7664	13.57 (13.29–13.86)	758	4.52 (4.21–4.85)	8597	7.55 (7.39–7.70)	393	4.47 (4.05–4.92)
Cadre staff	4169	7.38 (7.17–7.60)	973	5.81 (5.46–6.17)	6567	5.76 (5.63–5.90)	1395	15.88 (15.12–16.66)
Workers	3982	7.05 (6.84–7.27)	435	2.60 (2.36–2.85)	2989	2.62 (2.53–2.72)	361	4.11 (3.70–4.55)
Students	3475	6.15 (5.96–6.36)	1186	7.08 (6.70–7.48)	8722	7.66 (7.50–7.81)	761	8.66 (8.08–9.27)
Migrant workers	2768	4.90 (4.73–5.08)	294	1.75 (1.56–1.97)	1374	1.21 (1.14–1.27)	152	1.73 (1.47–2.03)
Commercial service	2236	3.96 (3.80–4.12)	460	2.75 (2.50–3.00)	1850	1.62 (1.55–1.70)	78	0.89 (0.70–1.11)
Retired staff	2466	4.37 (4.20–4.53)	280	1.67 (1.48–1.88)	4363	3.83 (3.72–3.94)	819	9.32 (8.72–9.95)
Teachers	1419	2.51 (2.39–2.65)	466	2.78 (2.54–3.04)	4214	3.70 (3.59–3.81)	448	5.10 (4.65–5.58)
Scattered children	798	1.41 (1.32–1.51)	590	3.52 (3.25–3.81)	2710	2.38 (2.29–2.47)	154	1.75 (1.49–2.05)
Catering food industry	814	1.44 (1.34–1.54)	519	3.10 (2.84–3.37)	687	0.60 (0.56–0.65)	14	0.16 (0.09–0.27)
Medical staff	479	0.85 (0.77–0.93)	143	0.85 (0.72–1.00)	1029	0.90 (0.85–0.96)	131	1.49 (1.25–1.77)
Herder	223	0.39 (0.34–0.45)	61	0.36 (0.28–0.47)	354	0.31 (0.28–0.34)	338	3.85 (3.46–4.27)
Child care	203	0.32 (0.28–0.37)	121	0.72 (0.60–0.86)	926	0.81 (0.76–0.87)	24	0.27 (0.18–0.41)
Others & unknown	2661	4.71 (4.54–4.89)	570	3.40 (3.13–3.69)	2539	2.23 (2.14–2.32)	103	1.17 (0.96–1.42)

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
