# Peer review of "Age-Period-Cohort Analysis on the Time Trend of Hepatitis B Incidence in Four Prefectures of Southern Xinjiang, China from 2005 to 2017"

_ijerph, 2019, doi:10.3390/ijerph16203886_

Round 1

Reviewer 1 Report

This is an interesting study that analyzes the change of incidence of hepatitis B in a closed cohort. The authors had developed a strong analysis framework based on the Age-Period-Cohort model for population statistical data of southern Xinjiang, China. The results provide a better understanding of epidemiology features and the relation between social events and the incidence of hepatitis B. 

Generally, the whole manuscript is well prepared. There are a few questions for the authors:

The change of population in Xinjiang should be considered. As there are several significant social-political policies enforced in Xinjiang, the population in Xinjiang might not static. There are two particular surges of incidence of hepatitis B in newborns in Aksu. Is there any explanation for it? There are too many figures, the authors may consider making their figure more condense for a better reading experience.

Author Response

Comments from the editors and reviewers: Reviewer 1 comments: 1. The change of population in Xinjiang should be considered. As there are several significant social-political policies enforced in Xinjiang, the population in Xinjiang might not static. There are two particular surges of incidence of hepatitis B in newborns in Aksu. Is there any explanation for it? Response: Thank you for the concerns and suggestions. Actually, annual population data of the four regions we use are from Xinjiang Statistical Yearbook 2005-2017. Unfortunately, we did not have detailed age structure each year. We applied the detailed age structure of the 2010 census data to the annual data of 2005-2017, combined the two groups of population data, and constructed the complete population age statistics of the four regions from 2005 to 2017. In figure 7, there are two particular surges of incidence of hepatitis B in 0-year-old infants in Aksu. In 2012, the hepatitis B cases were 33 and the number of cases in the rest years fluctuates around 10. We have called the staff of CDC in Aksu, they did not know the specific reason so far. 2. There are too many figures, the authors may consider making their figure more condense for a better reading experience. Response: We agree with the reviewer that We have updated the figs for a better reading experience.

Reviewer 2 Report

AUTHORS

Manuscript ID: ijerph-607093

Title: Age-period-cohort analysis on the time trend of Hepatitis B incidence in four Prefectures of Southern Xinjiang, China from 2005 to 2017

Line 21-22, start of Introduction: “Hepatitis B (HB) is an infectious disease with nflammatory lesions of the liver mainly caused by Hepatitis B virus (HBV) infection.” Please rephrase for clarity.

Between line 21 and line 30 I believe that text would gain by including explanation on HBV transmission, including in China.

Line 36-37: “It is also a big health problem priority to  be solved in the process of building a people-oriented harmonious society.” I believe it can be removed.

Line 84: “Meanwhile, we retrieved the relatively detailed age structure of the four regions (0, 1-4, 5-9, 10-14, ..., 85 95-99, and over 100)” can maybe be changed to “5-year interval categories”?

In materials and Methods, it is not obvious the amount of data available. I would recommend disclosing the number of people included in the study (before and after exclusion off > 100 year old).

Please reference and explain HB incidence in 100+ year old is negligible and below 0.001%

I feel the need to clarify how a HB case was defined in each of the databases used for retrieval of information. Are case definitions different?

I am not a vaccinologist but If case definitions are dependent on the detection of HB antibodies and vaccines were introduced, couldn’t there be reporting of vaccine antibodies from a time point onwards, ultimately being mistaken with cases of HB? Or is HB vaccine marked and for that reason distinctions can be made using serology? This needs to be clarified

Author Response

Reviewer 2 Comments: 1. line 21-22, start of Introduction: “Hepatitis B (HB) is an infectious disease mainly caused by Hepatitis B virus (HBV).” Please rephrase for clarity. Response: We have added the following sentence: “Hepatitis B (HB) is an inflammatory lesions of the liver mainly caused by hepatitis B virus (HBV), which is a kind of infectious disease as well.” 2. Between line 21 and line 30 I believe that text would gain by including explanation on HBV transmission, including in China. Response: We agree with the reviewer that we have added the following sentence to explain the HBV transmission in China: “HBV can be transmitted by parenteral, sexual, vertical and horizontal routes. Thus, individuals who engage in risky behaviors, such as the use of psychoactive substances, unprotected sex, multiple sexual partners, early initiation of sexual activity, and sharing of personal objects, are more vulnerable to HBV infection” 3. line 36-37: “It is also a big health problem priority to be solved in the process of building a people-oriented harmonious society.” I believe it can be removed. Response: We have removed the sentence “It is also a big health problem priority to be solved in the process of building a people-oriented harmonious society”. 4. Line 84: “Meanwhile, we retrieved the relatively detailed age structure of the four regions (0, 1-4, 5-9, 10-14, ..., 85 95-99, and over 100)” can maybe be changed to “5-year interval categories”? Response: Thank you for the concern. We are sorry that this age interval cannot be changed easily, because this data comes from China's 2010 census, in which the population age interval is (0 years old, 1-4 years old, 5-9 years old, 10-14 years old..., 85, 95-99, and over 100). 5. In materials and Methods, it is not obvious the amount of data available. I would recommend disclosing the number of people included in the study (before and after exclusion off > 100 year old). Please reference and explain HB incidence in 100+ year old is negligible and below 0.001%,I feel the need to clarify how a HB case was defined in each of the databases used for retrieval of information. Are case definitions different? Response: Thank you for the concern. We have clarified the number of data collected and the number of cases included in the study are detailed in the data source. There are 215413 HB cases, and the cases in four regions are: 63697 cases in Aksu, 18061 cases in Hotan, 124486 cases in Kashgar, and 9169 cases in Kizilsu kirghiz. We browsed, reviewed and downloaded the case sheet of HB reported in the information management system of infectious disease report from 2005 to 2018 according to the date of onset. Case table includes basic information, diagnosis type, diagnosis time and so on. There are only 17 cases (generally less than 0.001%), we deleted cases over 100 years old for convenience on data processing and exploration. 6. I am not a vaccinologist but If case definitions are dependent on the detection of HB antibodies and vaccines were introduced, couldn’t there be reporting of vaccine antibodies from a time point onwards, ultimately being mistaken with cases of HB? Or is HB vaccine marked and for that reason distinctions can be made using serology? This needs to be clarified. Response: Thank you for the concern. Hepatitis B surface antigen positive patients were diagnosed as HB cases. Surface antigen is positive is the index that judges a patient to infect, surface antibody is protective antibody, the crowd that surface antibody is positive won't infect second liver, surface antibody and surface antigen won't be positive at the same time.